# COVID-19 Immunisation, Willingness to Be Vaccinated and Vaccination Strategies to Improve Vaccine Uptake in Australia

**DOI:** 10.3390/vaccines9121467

**Published:** 2021-12-11

**Authors:** Bing Wang, Rebecca Nolan, Helen Marshall

**Affiliations:** 1Vaccinology and Immunology Research Trials Unit, Women’s and Children’s Health Network, Adelaide, SA 5006, Australia; bing.wang@adelaide.edu.au; 2Robinson Research Institute and Adelaide Medical School, The University of Adelaide, Adelaide, SA 5006, Australia; 3Epidemiology Branch, Prevention and Population Health Directorate, Wellbeing SA, Adelaide, SA 5006, Australia; rebecca.nolan@sa.gov.au

**Keywords:** COVID-19 vaccines, vaccine uptake, vaccine intention, vaccine policies

## Abstract

The COVID-19 vaccine rollout is crucial to lifting community and economic restrictions. This cross-sectional study aimed to assess: (a) COVID-19 vaccine uptake and associated factors; (b) COVID-19 vaccine intentions and associated factors; (c) community support for COVID-19 vaccination strategies and associated factors. The survey was conducted between May and July 2021 in Australia. Of 3003 participants, 30% reported they were already vaccinated and 39% indicated they would get vaccinated. Low socioeconomic and education levels, non-English speaking backgrounds and being parents were associated with decreased vaccine willingness and/or vaccination rates. High levels of support for vaccination strategies were demonstrated with mandatory vaccination being less preferable. Respondents from non-English speaking backgrounds were more likely to support a mandatory vaccination policy. Respondents with the highest socioeconomic level were more likely to support vaccination requirements for international travel, visiting nursing homes and working in healthcare settings. Respondents who were aged ≥70 years were more likely to support all proposed vaccination strategies. Targeted campaigns should be implemented for parents and those who live in socioeconomic disadvantaged areas and have lower educational attainment. Concise and clear vaccine information should be provided in lay and multiple languages to improve vaccine confidence. Vaccine enforcement policies should be considered and implemented with caution.

## 1. Introduction

COVID-19 vaccines have been approved and rolled out in many countries. In Australia, it has been estimated that 85% of the Australian population including 5–16 year-olds will have to be vaccinated to achieve herd immunity [1]. Initial shortage of vaccine supply, delays in local production and public concerns over the safety of COVID-19 vaccines with intensive media coverage have impacted Australia’s rollout. Although 47.5% of the world population has received at least one dose of a COVID-19 vaccine [2], concerns about vaccination hesitancy had been raised even prior to the start of rollouts and has been reported worldwide [3,4]. High vaccination coverage is critical to lifting COVID-19 restrictions and achieving potential herd immunity. There is an urgent need to examine vaccine uptake and public opinion towards vaccination policies in order to develop evidence-based communication strategies to build trust and continue to improve COVID-19 vaccine uptake in target groups. 

It is likely that population attitudes towards the COVID-19 vaccine will fluctuate with further waves of the pandemic, necessitating regular tracking of vaccine confidence among different population groups to ensure public health campaigns remain responsive to community vaccine sentiments. In a longitudinal survey conducted in Australia between January and August 2021, there were several sub-populations identified that remained hesitant about COVID-19 vaccination and had lower vaccination rates [5,6]. Various vaccine campaign strategies have been considered or implemented by the Government or private corporations. For example, COVID-19 vaccines have become mandatory for aged care workers and most people working on the quarantine front line in Australia. Shepparton food processor SPC became the first company in Australia to mandate COVID-19 vaccination of all staff [7]. However, very few studies have been conducted to explore community acceptance of those vaccination enforcement policies and campaign strategies. More children and young people have been infected with the Delta variant due to high transmissibility and as they remain an unvaccinated group. The COVID-19 vaccine rollout has been extended to children aged between 12 and 15 years in many countries including the US, Australia and other European countries. However, parental support for their children to be vaccinated has not been well captured [6,8,9].

Our study aimed to assess: (1) COVID-19 vaccine uptake; (2) sociodemographic and health factors associated with vaccine uptake; (2) vaccine willingness/hesitancy to immunise themselves/their child; (4) sociodemographic and health factors associated with vaccine intentions; (5) community support for Government strategies to improve COVID-19 uptake; (6) sociodemographic and health factors associated with support for COVID-19 vaccination strategies.

## 2. Materials and Methods

### 2.1. Study Design and Data Sources

A cross-sectional survey design was implemented through the Population Health Survey Module System (PHSMS). The PHSMS is an ‘omnibus-type’ service available to government and non-government organisations to obtain data on a range of population health and wellbeing issues within South Australia, Australia. 

### 2.2. Data Collection

Data were collected from South Australian adults aged 18 years and over using a computer-assisted telephone interviewing system. The cooperation rate was 77.3% [10]. The survey questions (Appendix A) were approved by the Department for Health and Wellbeing Human Research Ethics Committee (HREC/18/SAH/78).

### 2.3. Statistical Analysis

Data are presented by various population sub-groups of sociodemographic and health characteristics. The categorisation criteria are described in the Appendix A. Sociodemographic variables such as gender, age, Aboriginal and Torres Strait Islander status, culturally and linguistically diverse backgrounds, marital status, socioeconomic status (SES), education attainment and employment status, were summarised with mean values and standard deviations for continuous variables and percentages and 95% confidence intervals for categorical variables. Respondents from culturally and linguistically diverse backgrounds were measured using the respondent’s country of birth (COB): English speaking countries and non-English speaking countries. Socioeconomic status was assessed by quintiles of postcode-based Australian Socio-Economic Index For Areas (SEIFA) according to the definition from the Australian Bureau of Statistics (ABS) [11]. 

The predictor variables comprised of age, gender, Aboriginality, SEIFA, marital status, educational attainment, employment status, country of birth, metropolitan/country areas and presence of chronic medical conditions. Parental status was an additional predictor variable in the regression model to investigate factors associated with COVID-19 vaccination rates. Respondents’ vaccination status and willingness to vaccinate themselves was an additional predictor variable in the model assessing parental willingness to vaccinate their children. The outcome measures included vaccine willingness to get vaccinated for themselves or their children, self-reported vaccination status, and support for mandatory vaccination, vaccination reminders and vaccine requirements for international and domestic travel, visiting nursing homes and working in healthcare settings. When ordered logistic regression models did not demonstrate adequate fit, logistic regression models were used. The outcome was defined as 1 if a respondent answered “very likely/strongly agree” and 0 for any other responses in the logistic regression models. Logistic regression analyses were performed for vaccination status and travel requirements.

Univariate and multivariate logistic regression analyses were performed to test association between predictor variables and outcome measures. All multivariate logistic regression models only included covariates that achieved bivariate regression threshold of statistical significance (*p* ≤ 0.20) in the univariate logistic regress analyses. All results presented in the regression analyses were weighted. Raking was used to weight respondents incorporating various population characteristics to more closely reflect the South Australian population using benchmarks derived from the June 2016 ABS Census data. With raking, we chose a set of variables (gender, age, area of residence, country of birth, dwelling status, marital status, education level, employment status, household size) where the South Australian population distribution is known, and the raking procedure iteratively adjusts the weight for each survey participant until the sample distribution aligns with the South Australian population [12]. The weighting process ensured our results were representative of the South Australian population as a whole.

## 3. Results

### 3.1. Study Population

This survey was conducted with 3003 South Australians between May and July 2021 including 718 parents or caregivers. Weighted data show approximately equal number of males and females and 2.4% of those interviewed being Aboriginal or Torres Strait Islander (Table 1). In comparison with the Australian adult population, our survey respondents are generally representative with the exception of South Australians being older, less culturally diverse and more likely to be single.

### 3.2. Factors Associated with COVID-19 Vaccination

Of 3003 respondents, 30.0% reported that they were already vaccinated (Table 2). In the weighted and adjusted logistic regression, vaccination rates were lower in younger adults, those without a chronic condition, those from non-English speaking backgrounds, those from lower socioeconomic areas and those with an education level of <Year 12 (Table 3). Parents or caregivers of a child/children aged <16 years were less likely to have been vaccinated than their counterparts (aOR = 0.65, 95%CI: 0.50–0.86).

### 3.3. Factors Associated with Willingness to Receive COVID-19 Vaccine

Of 3003 respondents, 39.3% indicated they would agree to be vaccinated (Table 2). Exactly half of parents or caregivers of a child/children aged less than 16 years indicated that they were very likely to get their children vaccinated (50.5%, 95%CI: 46.4%–54.7%). However, 8.1% (95%CI: 7.0%–9.4%) of respondents stated that they would not be vaccinated and 13.5% (95%CI: 10.6%–16.9%) of parents indicated that they were not at all likely to get their child vaccinated.

After excluding vaccinated respondents and those who preferred not to say from the weighted and adjusted ordered logistic regression analysis, respondents who had the lowest SES, were not married and had <Year 12 education, were less willing to be vaccinated with the COVID-19 vaccine compared with those who had the highest SES, were married and had ≥Year 12 education (Table 4). A higher proportion of Aboriginal and Torres Strait Islander people were undecided about COVID-19 vaccination (49.6%, 95%CI: 34.0%–65.3%) than non-Indigenous people (31.2%, 95%CI: 28.8%–33.7%) and less stated they would be vaccinated (37.5% (95%CI: 23.5%–53.8%) vs. 57.1% (95%CI: 54.5%–59.7%)). 

After controlling for parents’ vaccination status and willingness and other sociodemographic factors in the ordered logistic regression (Appendix A), those who indicated that they have been vaccinated or will be vaccinated significantly increased the likelihood of having their children vaccinated (aOR: 14.10, 95%CI: 9.15–21.71). 

### 3.4. Public Support of Potential Vaccination Policies and Strategies

Most respondents (84.8%, 95%CI: 83.2%–86.2%) supported the vaccination requirements for international travel and 60.7% (95%CI: 58.7%–62.7%) for domestic travel. Proof of vaccination was strongly supported by 67.9% (95%CI: 66.0%–69.9%) for visiting Residential Aged Care Homes and working in healthcare settings. Only 30.1% (95%CI: 28.2%–31.9%) strongly agreed that the Government should make COVID-19 vaccination mandatory. Vaccination reminders were strongly supported by 46.7% (95%CI: 44.7%–48.8%) of respondents (Table 5). Those aged ≥70 years were more likely to support all those proposed vaccination policies and strategies (Appendix A). Respondents from non-English backgrounds were more likely to support mandatory vaccination policy (Appendix A). Respondents with the highest SES were more likely to support the vaccination requirements for international travel, visiting Residential Aged Care Homes and working in healthcare settings (Appendix A). Chronic illness was associated with a higher level of vaccination support for domestic travel (Appendix A). Women were less willing to support the mandatory vaccination policy (Appendix A).

## 4. Discussion

The majority of respondents indicated they were very likely to receive the COVID-19 vaccine, or they were already vaccinated with at least one dose of a COVID-19 vaccine. However, 8% were resistant to vaccination and 14% were not at all likely to get their child vaccinated. This will result in a vulnerable unvaccinated group when COVID-19 restrictions are eased in Australia. Vaccine willingness and resistance have been investigated worldwide using different methodologies. Intention to receive the COVID-19 vaccine has fluctuated over time and between countries [3,4,5,14,15,16,17,18,19,20,21,22]. A large decline (15%) in vaccine willingness between August 2020 and January 2021 was observed in an Australian national longitudinal survey [16]. In Australia, vaccine hesitancy and lower vaccination rates were observed in a few sub-populations [5,6]. In a parent online survey conducted in June 2020, 24% of Australian parents were unsure or unwilling to accept a COVID-19 vaccine [23]. Around one third of parents reported that they were unsure (don’t know) or not likely to get their children vaccinated in our survey, which was lower than the proportion of parental vaccine acceptance reported in other surveys conducted in the UK and Australia [6,8]. In the UK, a higher level of vaccine acceptance for adults than for their child/children was reported, which was similar to our study results [8]. Our results showed that parents were less likely to have their children vaccinated if they were not willing to get vaccinated or have not been vaccinated themselves. The comparable findings were reported in Japan that parents’ willingness to get vaccinated for themselves was a significant factor for parental willingness to get children vaccinated [9]. Moreover, after adjusting for age and other sociodemographic factors, being a parent was associated with a lower likelihood of being vaccinated.

Social, political, health and demographic factors associated with intention to receive a COVID-19 vaccine have been investigated and results vary between countries, studies and survey times. In line with a few other surveys [3,5,16,17,21,23,24,25,26,27,28], our survey showed the lower education level and socioeconomic disadvantage were associated with reduced vaccine willingness and vaccination uptake. Association between age and vaccine willingness was inconsistently reported in previous surveys [3,5,16,17,18,23,24,25,28,29,30]. Similar to a previous survey [17], age and chronic illness were not associated with vaccine willingness in our study. However, older respondents with chronic illness were more likely to have already received the vaccine, which may reflect that the COVID-19 vaccine rollout in Australia gradually expanded and elderly people and people with underlying medical conditions were prioritised to receive the vaccine in the earlier phase of the rollout. Factors associated with vaccination uptake were assessed in respondents who self-reported that they had received at least one dose of vaccination in surveys conducted in April (9%) and August (57%) 2021. In line with the above mentioned surveys [5,6], it was noted that educational attainment and socioeconomic status were associated with vaccination rates. Respondents from non-English speaking backgrounds were less likely to have been vaccinated although there was no difference in vaccine willingness compared to those of English speaking background [5,6,16]. Despite our small sample size, Aboriginal and Torres Strait Islander people are more hesitant to receive COVID-19 vaccines. Further research led by Aboriginal and Torres Strait Islander people would improve our understanding of factors associated with COVID-19 vaccine hesitancy in this important group. 

This study found that respondents were generally supportive of mandating vaccination in some high risk settings, for the purpose of travelling and reminder strategies. The high level of support of vaccination requirements for international travel and a lower level of support for domestic travel is consistent with concerns about risk to Australians of overseas exposure of the COVID-19 variants. Although there is majority community support for these measures, there are still exemptions, and legal, ethical and technological issues to consider [31,32]. An equally high proportion of respondents agreed that COVID-19 vaccination should be required for visiting nursing homes and working in healthcare settings to reduce the risk to vulnerable people. In the US, a case study demonstrated that a COVID-19 vaccination condition of employment should be implemented with caution, as it required efforts and involvement from staff at all levels of the workplace, adequate time for developing and disseminating information, transparent communication and sufficient funding [33]. Compared to a previous survey reporting that 73% agreed that the Government should require a COVID-19 vaccine for work, travel and study in 2020, our survey showed a lower level of support for possible Government mandates (54% strongly or somewhat agreed). Similar to the survey conducted in 2020, older respondents and higher income respondents were more likely to support mandatory vaccination [14]. Although mandatory vaccination strategies may be a practical solution to improve vaccine uptake within a short period and effectiveness has been demonstrated in the childhood immunisation program and several other contexts [34], studies have found that vaccine enforcement reduced overall vaccine acceptance which may potentially undermine the goals of COVID-19 vaccination campaigns in the US [20,35]. Simple reminder messages, letters or emails have increased uptake of seasonal flu vaccines [36], and uptake at the beginning of the COVID-19 vaccine rollout in the US [37]. However, a randomised controlled trial found short messages did not increase vaccination rates in the US [38], which may indicate text messages were more useful to turn pre-existing intentions into behaviours rather than convincing people with vaccine hesitancy. This strategy was supported by most respondents.

The survey was administered in a representative weighted sample with one third reporting that they had received at least one vaccination dose. Although respondents self-reported their vaccination status, we expect minimal recall bias due to the recent onset of the COVID-19 vaccine program. Comparisons with other surveys over time and in other countries should be interpreted with caution, as public attitudes change and evolve in response to important media announcements and major events. Our findings are limited to the cross-sectional design, being one snapshot in time but are current to the situation in Australia of outbreaks and lockdowns. The study sample is representative of the South Australian population, and may not be generalised to the whole Australian population which is slightly younger (median age: 38 years (Australian population); 40 years (South Australian population)) and more culturally diverse (percentage of people were born in Australia: 66.7% (Australian population); 71.1% (South Australian population)) [13].

## 5. Conclusions

The COVID-19 vaccine rollout has already expanded to children aged 12 and older in Australia and other countries. The vaccine rollout will depend on widespread community trust of COVID-19 vaccines including parents. Despite enormous efforts that have delivered the COVID-19 vaccine rollout, governments, health professionals and COVID-19 vaccination providers may still need to address vaccine hesitancy in parents and other socioeconomically disadvantaged groups by providing evidence-based information in lay language for the limited-English-proficient populations, and by developing effective vaccine campaigns in languages other than English for migrants and refugees. Equity in protecting people with low SES and education or from non-English speaking backgrounds and Aboriginal and Torres strait Islander peoples will need extra resourcing to ensure everyone is adequately protected, and this will be of benefit to the whole community. The vaccination strategies to improve uptake in less willing sub-groups should be considered with caution. Future research is paramount to monitor vaccine willingness and uptake, and adjust communication strategies to improve vaccine uptake in people who were uncertain or less willing to be vaccinated.

## Figures and Tables

**Table 1 vaccines-09-01467-t001:** Weighted and unweighted sociodemographic characteristics of all respondents.

	**Weighted (*n* = 3003)**		**Unweighted (*n* = 3003)**		**Australia** [13]
	*n*	% (95%CI)	*n*	% (95%CI)	%
Gender					
Male	1460	48.7 (46.7–50.7)	1528	51.0 (49.2–52.8)	48.8
Female	1537	51.3 (49.3–53.3)	1469	49.0 (47.2–50.8)	51.2
Age (median)	49		55		46
18–29 years	584	19.5 (17.8–21.2)	480	16.0 (14.7–17.3)	20.9
30–49 years	949	31.6 (29.7–33.6)	798	26.6 (25.0–28.2)	35.3
50–69 years	927	30.9 (29.0–32.8)	1021	34.0 (32.3–35.7)	30.0
≥70 years	543	18.1 (16.7–19.5)	704	23.4 (22.0–25.0)	13.7
Aboriginal status					
No	2913	97.6 (96.8–98.2)	2932	98.2 (97.7–98.6)	97.7
Yes	71	2.4 (1.8–3.2)	53	1.8 (1.4–2.3)	2.3
SEIFA					
lowest quintile	555	18.5 (16.9–20.2)	494	16.5 (15.2–17.8)	
low quintile	663	22.1 (20.4–23.8)	617	20.6 (19.1–22.0)	
middle quintile	632	21.0 (19.4–22.7)	643	21.4 (20.0–22.9)	
high quintile	547	18.2 (16.7–19.9)	558	18.6 (17.2–20.0)	
highest quintile	606	20.2 (18.6–21.8)	690	23.0 (21.5–24.5)	
Marital status					
Single	1493	50.2 (48.2–52.3)	1733	58.2 (56.4–59.9)	49.7
Married/de facto	1480	49.8 (47.7–51.8)	1246	41.8 (40.1–43.6)	50.3
Education level					
Lower than Year 12 education	714	23.9 (22.2–25.7)	637	21.3 (19.9–22.8)	28.3
≥Y12/TAFE/certificate/diploma	1685	56.5 (54.5–58.5)	1311	43.8 (42.1–45.6)	48.7
Degree or higher	583	19.6 (18.3–20.9)	1042	34.8 (33.2–36.6)	23.0
Employment					
Unemployed	127	4.2 (3.6–5.0)	163	5.5 (4.7–6.3)	4.3
Employed ^	1654	55.4 (53.4–57.4)	1568	52.4 (50.6–54.2)	61.6
Others ^^	1205	40.4 (38.4–42.4)	1259	42.1 (40.3–43.9)	34.1
Household income					
<AUD 60,000	830	27.6 (25.8–29.5)	829	27.6 (26.0–29.2)	
AUD 60,000-AUD 150,000	789	26.3 (24.5–28.1)	803	26.7 (25.2–28.4)	
>AUD 150,000	303	10.1 (9.0–11.3)	345	11.5 (10.4–12.7)	
UNK	1081	36.0 (34.1–38.0)	1026	34.2 (32.5–35.9)	
COB (Country of Birth)					
COB English main language	2551	85.2 (83.8–86.6)	2480	82.9 (81.5–84.2)	77.0
COB Non-English speaking	443	14.8 (13.4–16.2)	512	17.1 (15.8–18.5)	23.0
Area of residence					
Metro. Adelaide	2175	72.4 (70.6–74.2)	2199	73.2 (71.6–74.8)	
SA Country	828	27.6 (25.8–29.4)	804	26.8 (25.2–28.4)	
Chronic medical conditions					
No	1554	51.9 (49.8–53.9)	1502	50.2 (48.4–51.9)	
Yes	1442	48.1 (46.1–50.2)	1493	49.9 (48.1–51.6)	
Number of children living in household					
0	2238	74.5 (72.7–76.2)	2250	75.3 (73.7–76.8)	
1	348	11.6 (10.3–13.0)	334	11.2 (10.1–12.4)	
≥2	395	13.3 (12.0–14.7)	404	13.5(12.3–14.8)	

Note: The weighting of the data can result in rounding discrepancies or totals not adding. CI: confidence interval. Missing data were excluded from descriptive analyses. ^ Employed included full-time employed, part-time employed and casual employment. ^^ Other category includes respondents engaged in home duties, unable to work, carers, volunteers, students, retired and any other response.

**Table 2 vaccines-09-01467-t002:** Descriptive statistics of self-reported vaccination status and vaccine willingness to get vaccinated for themselves or their children.

**Vaccine Willingness or Vaccination Status**	** *n* ** **/N (Weighted)**	**% (95%CI) (Weighted)**	** *n* ** **/N (Unweighted)**	**% (95%CI) (Unweighted)**
Which of the following statements about the COVID-19 vaccine best apply to you?				
I have been vaccinated (one or two doses)	901/3003	30.0 (28.3–31.8)	1107/3003	36.9 (35.2–38.6)
I will be getting vaccinated when it becomes available to me	1181/3003	39.3 (37.3–41.4)	1136/3003	37.8 (36.1–39.6)
I will not be getting vaccinated when it becomes available to me	243/3003	8.1 (7.0–9.4)	196/3003	6.5 (5.7–7.5)
I am undecided whether or not I will be getting vaccinated when it becomes available to me	660/3003	22.0 (20.3–23.8)	552/3003	18.4 (17.0–19.8)
Prefer not to say	18/3003	0.6 (0.3–1.2)	12/3003	0.4 (0.2–0.7)
If you are a parent or caregiver of a child/children aged less than 16 years, please answer this question. If the COVID-19 vaccine is safe, effective and approved to use in children by the Government, how likely would you be to get your child/children vaccinated?				
Very likely	362/717	50.5 (46.4–54.7)	387/718	53.9 (50.2–57.5)
Somewhat likely	145/717	20.3 (17.2–23.7)	144/718	20.1 (17.3–23.2)
Not very likely	57/717	8.0 (5.9–10.6)	56/718	7.8 (6.0–10.0)
Not at all likely	96/717	13.5 (10.6–16.9)	78/718	10.9 (8.8–13.4)
Don’t know	43/717	6.1 (4.4–8.3)	46/718	6.4 (4.8–8.5)
Prefer not to say	13/717	1.7 (0.8–4.0)	7/718	1.0 (0.5–2.0)

**Table 3 vaccines-09-01467-t003:** Adjusted odds ratios of receiving at least one dose of the COVID vaccine.

	Respondents Who Were Vaccinated with at Least One Dose of the COVID Vaccine		Multivariate Logistic Regression	
	*n*/N (weighted)	% (95%CI) (weighted)	aOdds Ratio	95%CI
Age				
18–29years	60/582	10.3 (7.8–13.5)	Ref *	
30–49years	167/939	17.8 (15.1–21.0)	1.75	1.16–2.65
50–69years	329/922	35.7 (32.5–39.1)	3.89	2.68–5.65
≥70years	344/542	63.6 (59.6–67.4)	13.55	8.85–20.75
Gender				
Male	401/1445	27.8 (25.4–30.3)	Ref *	
Female	500/1534	32.6 (30.1–35.2)	1.09	0.89–1.33
Aboriginal status				
No	883/2898	30.5 (28.7–32.3)	Ref *	
Yes	13/71	18.7 (9.5–33.4)	1.01	0.39–2.60
SEIFA				
lowest quintile	127/549	23.2 (19.5–27.3)	Ref *	
low quintile	202/658	30.8 (27.0–34.7)	1.31	0.95–1.80
middle quintile	201/630	31.8 (28.0–35.9)	1.60	1.16–2.21
high quintile	150/544	27.5 (23.7–31.7)	1.19	0.85–1.66
highest quintile	221/603	36.7 (32.7–40.8)	1.76	1.27–2.44
Marital status				
Married/de facto	499/1489	33.5 (31.2–36.0)	Ref *	
Single	397/1465	27.1 (24.5–29.8)	0.82	0.67–1.01
Education level				
Lower than Year 12 education	246/706	34.8 (31.0–38.9)	Ref *	
≥Year 12/TAFE/certificate/diploma	439/1679	26.1 (23.8–28.7)	1.09	0.85–1.39
Degree or higher	215/582	37.0 (33.9–40.2)	2.24	1.69–2.97
Employment				
Unemployed	20/124	16.1 (11.1–22.8)	Ref *	
Employed ^	378/1648	23.0 (20.8–25.3)	1.36	0.83–2.23
Others ^^	498/1196	41.6 (38.6–44.7)	1.25	0.75–2.10
COB (Country of Birth)				
COB English main language	798/2533	31.5 (29.6–33.5)	Ref *	
COB Non-English speaking	101/441	22.9 (19.1–27.2)	0.72	0.54–0.95
Area of residence ^				
Metro. Adelaide	617/2158	28.6 (26.6–30.7)	Ref *	
SA Country	284/827	34.3 (30.9–38.0)	1.25	0.99–1.57
Chronic medical conditions				
No	342/1544	22.1 (20.0–24.4)	Ref *	
Yes	556/1434	38.8 (36.1–41.5)	1.27	1.03–1.57
Parents or caregivers of a child/children aged <16 years				
No	770/2277	33.8 (31.7–36.0)	Ref *	
Yes	131/708	18.5 (15.7–21.6)	0.65	0.50–0.86

^ Employed included full-time employed, part-time employed and casual employment. ^^ Other category includes respondents engaged in home duties, unable to work, carers, volunteers, students, retired and any other response. * Reference group

**Table 4 vaccines-09-01467-t004:** Weighted and adjusted odds ratios of likelihood of taking the COVID vaccine themselves.

	Respondents Who Stated “I Will Not Be Getting Vaccinated”		Respondents Who Stated “I Am Undecided”		Respondents Who Stated “I Will Be Getting Vaccinated”		Multivariate Ordered Logistic Regression	
	*n*/N (weighted)	% (95%CI) (weighted)	*n*/N (weighted)	% (95%CI) (weighted)	*n*/N (weighted)	% (95%CI) (weighted)	aOdds Ratio	95%CI
Age								
18–29 years	45/522	8.6 (6.1–12.0)	173/522	33.1 (28.4–38.2)	304/522	58.3 (53.0–63.3)		
30–49 years	89/771	11.5 (8.9–14.8)	246/771	31.9 (27.9–36.1)	437/771	56.6 (52.2–60.9)		
50–69 years	83/593	14.0 (11.0–17.8)	189/593	31.8 (27.6–36.4)	321/593	54.1 (49.4–58.8)		
≥70 years	26/197	13.0 (8.7–19.0)	52/197	26.6 (20.9–33.2)	119/197	60.4 (53.4–67.0)		
Gender								
Male	117/1044	11.2 (9.1–13.7)	327/1044	31.3 (28.1–34.7)	600/1044	57.5 (53.9–60.9)		
Female	124/1034	12.0 (9.7–14.7)	333/1034	32.2 (28.8–35.8)	577/1034	55.8 (52.1–59.5)		
Aboriginal status								
No	235/2015	11.7 (10.0–13.5)	628/2015	31.2 (28.8–33.7)	1151/2015	57.1 (54.5–59.7)	Ref *	
Yes	7/58	12.9 (5.5–27.6)	29/58	49.6 (34.0–65.3)	22/58	37.5 (23.5–53.8)	0.85	0.45–1.59
SEIFA								
lowest quintile	65/422	15.4 (11.6–20.2)	146/422	34.6 (29.3–40.4)	211/422	49.9 (44.2–55.7)	Ref *	
low quintile	63/455	13.8 (10.4–18.1)	153/455	33.7 (28.5–39.2)	239/455	52.6 (47.0–58.1)	1.02	0.74–1.40
middle quintile	42/430	9.9 (6.9–13.8)	132/430	30.8 (25.8–36.2)	255/430	59.4 (53.8–64.7)	1.24	0.90–1.72
high quintile	41/395	10.5 (7.1–15.3)	123/395	31.3 (26.1–37.0)	230/395	58.2 (52.3–64.0)	1.16	0.82–1.63
highest quintile	31/382	8.2 (5.4–12.1)	105/382	27.5 (22.6–33.0)	245/382	64.3 (58.6–69.7)	1.45	1.04–2.03
Marital status								
Married/de facto	87/989	8.8 (7.0–11.1)	297/989	30.0 (27.0–33.3)	605/989	61.1 (57.7–64.4)	Ref *	
Single	151/1069	14.2 (11.7–17.1)	354/1069	33.1 (29.6–36.8)	564/1069	52.7 (48.9–56.5)	0.74	0.60–0.91
Education level								
Lower than Year 12 education	77/460	16.7 (12.9–21.3)	183/460	39.7 (34.4–45.3)	201/460	43.6 (38.3–49.1)	Ref *	
≥Year 12/TAFE/certificate/diploma	141/1240	11.4 (9.3–13.9)	396/1240	32.0 (28.8–35.3)	703/1240	56.7 (53.1–60.1)	1.59	1.24–2.05
Degree or higher	18/104	6.6 (4.6-,9.28)	41/104	20.3 (17.1–23.9)	46/104	73.2 (69.2–76.8)	2.91	2.14–3.96
Employment								
Unemployed	134/1269	17.2 (11.2–25.3)	386/1269	39.0 (30.3–48.4)	749/1269	43.9 (34.9–53.3)	Ref *	
Employed ^	208/1736	10.6 (8.6–12.9)	550/1736	30.4 (27.4–33.6)	978/1736	59.0 (55.7–62.3)	1.42	0.95–2.11
Others ^^	34/340	12.8 (10.1–16.3)	108/340	33.0 (28.9–37.4)	198/340	54.2 (49.7–58.6)	1.48	0.98–2.24
COB (Country of Birth)								
COB English main language	208/1736	12.0 (10.2–14.0)	550/1736	31.7 (29.1–34.4)	978/1736	56.3 (53.5–59.1)		
COB Non-English speaking	34/340	9.9 (6.4–15.0)	108/340	31.9 (26.6–37.7)	198/340	58.2 (52.2–64.0)		
Area of residence ^								
Metro. Adelaide	176/1541	11.4 (9.6–13.5)	476/1541	30.9 (28.2–33.8)	889/1541	57.7 (54.7–60.6)		
SA Country	67/543	12.3 (9.3–16.2)	184/543	33.9 (29.3–38.8)	292/543	53.8 (48.8–58.7)		
Chronic medical conditions								
No	122/1202	10.2 (8.3–12.5)	373/1202	31.0 (28.0–34.2)	707/1202	58.8 (55.5–62.1)	Ref *	
Yes	120/878	13.7 (11.1–16.8)	287/878	32.7 (29.0–36.5)	471/878	53.6 (49.7–57.5)	0.92	0.74–1.14

^ Employed included full-time employed, part-time employed and casual employment. ^^ Other category includes respondents engaged in home duties, unable to work, carers, volunteers, students, retired and any other response. * Reference group

**Table 5 vaccines-09-01467-t005:** Descriptive statistics of respondents who supported potential vaccination policies and strategies.

**Potential Vaccination Policies and Strategies**	** *n* ** **/N (Weighted)**	**% (95%CI) (Weighted)**	** *n* ** **/N (Unweighted)**	**% (95%CI) (Unweighted)**
Do you think COVID-19 vaccination or proof of vaccination should be required for: (Select all that apply)				
International travelling	2546/3003	84.8 (83.2–86.2)	2600/3003	86.6 (85.3–87.8)
Domestic travelling	1824/3003	60.7 (58.7–62.7)	1907/3003	63.5 (61.8–65.2)
Should not be required	294/3003	9.8 (8.5–11.2)	246/3003	8.2 (7.3–9.2)
Don’t know	137/3003	4.6 (3.8–5.5)	134/3003	4.5 (3.8–5.3)
Prefer not to say	10/3003	0.3 (0.1–0.7)	6/3003	0.2 (0.1–0.4)
COVID-19 vaccination or proof of vaccination should be required for visiting Residential Aged Care Homes and working in a hospital or healthcare clinic.				
Strongly Agree	2040/3003	67.9 (66.0–69.9)	2120/3003	70.6 (68.9–72.2)
Somewhat agree	509/3003	17.0 (15.5–18.6)	493/3003	16.4 (15.1–17.8)
Neither agree or disagree	107/3003	3.6 (2.9–4.4)	100/3003	3.3 (2.7–4.0)
Somewhat disagree	163/3003	5.4 (4.5–6.5)	139/3003	4.6 (3.9–5.4)
Strongly disagree	132/3003	4.4 (3.6–5.4)	110/3003	3.7 (3.0–4.4)
Don’t know	37/3003	1.2 (0.8–1.8)	30/3003	1.0 (0.7–1.4)
Prefer not to say	14/3003	0.5 (0.2–0.9)	11/3003	0.4 (0.2–0.7)
COVID-19 vaccination should be made mandatory by the Government.				
Strongly Agree	903/3003	30.1 (28.2–31.9)	962/3003	32.0 (30.4–33.7)
Somewhat agree	725/3003	24.1 (22.5–25.9)	756/3003	25.2 (23.7–26.8)
Neither agree or disagree	261/3003	8.7 (7.6–9.9)	261/3003	8.7 (7.7–9.8)
Somewhat disagree	468/3003	15.6 (14.2–17.1)	463/3003	15.4 (14.2–16.8)
Strongly disagree	561/3003	18.7 (17.1–20.4)	484/3003	16.1 (14.8–17.5)
Don’t know	74/3003	2.5 (1.9–3.2)	68/3003	2.3 (1.8–2.9)
Prefer not to say	12/3003	0.4 (0.2–0.8)	9/3003	0.3 (0.2–0.6)
Tailored vaccine reminder message/letter should be sent to everyone.				
Strongly Agree	1403/3003	46.7 (44.7–48.8)	1479/3003	49.3 (47.5–51.0)
Somewhat agree	858/3003	28.6 (26.7–30.5)	827/3003	27.5 (26.0–29.2)
Neither agree or disagree	219/3003	7.3 (6.3–8.4)	215/3003	7.2 (6.3–8.1)
Somewhat disagree	261/3003	8.7 (7.6–9.9)	250/3003	8.3 (7.4–9.4)
Strongly disagree	202/3003	6.7 (5.7–7.9)	179/3003	6.0 (5.2–6.9)
Don’t know	52/3003	1.7 (1.3–2.4)	47/3003	1.6 (1.2–2.1)
Prefer not to say	8/3003	0.3 (0.1–0.7)	6/3003	0.2 (0.1–0.4)

## Data Availability

The data presented in this study are available from the corresponding author (H.M.) upon reasonable request and final ethics approval.

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
