# Peer review of "COVID-19 Immunisation, Willingness to Be Vaccinated and Vaccination Strategies to Improve Vaccine Uptake in Australia"

_vaccines, 2021, doi:10.3390/vaccines9121467_

Round 1

Reviewer 1 Report

This paper has some merits and is well qualified for a potential publication.

Nonetheless, I would like that, at least, the following issue be better explained/discussed.

Essentially, here we are with a "characterization" of people who wish/do not wish get the vaccination in Australia.

As this is a very delicate and crucial point, I would be sure that:

  • this research question be better and more clearly declared at the beginning of the paper (abstract, Intro)
  • the sample of people used to conduct this research be really representative of the Australian society
  • the statistical method/techniques used to process the data be adequate to this task (as to this point, I am not that confident that a logistic regression be the more adequate choice; for example, given the sample used, maybe some bayesian techniques would be more favorite on my side, like those employed in these two studies:                        Reopening Italy’s schools in September 2020: a Bayesian estimation of the change in the growth rate of new SARS-CoV-2 cases, BMJ Open http://dx.doi.org/10.1136/bmjopen-2021-051458; 

    Modeling COVID-19 pandemic using Bayesian analysis with application to Slovene data, Mathematica Bioscieces, 2020, https://doi.org/10.1016/j.mbs.2020.108466

    ) .

Obviously, I am not asking to revoluzionize the entire paper but some more reflections and attention to the issues above.

Author Response

This paper has some merits and is well qualified for a potential publication.

Response: Thank you.

Nonetheless, I would like that, at least, the following issue be better explained/discussed.

Essentially, here we are with a "characterization" of people who wish/do not wish get the vaccination in Australia.

As this is a very delicate and crucial point, I would be sure that:

  • this research question be better and more clearly declared at the beginning of the paper (abstract, Intro)

Response: Revised.

Abstract: Vaccine willingness, vaccination status, acceptance of vaccination strategies to improve vaccine coverage, and associated factors were investigated in this cross-sectional survey. This cross-sectional study aimed to assess COVID-19 vaccine uptake and intentions, and acceptance toward certain vaccine policies. Individual characteristics associated with having been vaccinated/ intending to be vaccinated themselves/their child, and supporting vaccination enforcement strategies were investigated.

Introduction: Our study aimed to assess sociodemographic and health factors associated with COVID-19 immunisation and willingness and hesitancy to be vaccinated. Community support for Government strategies to improve COVID-19 uptake was also assessed. Our study aimed to assess COVID-19 vaccine uptake and willingness/hesitancy to immunise themselves/their child. Community support for Government strategies to improve COVID-19 uptake was assessed. Individual characteristics associated with having been vaccinated/ intending to be vaccinated themselves/their child, and supporting vaccination enforcement strategies were also investigated.

  • the sample of people used to conduct this research be really representative of the Australian society

Response: South Australia’s population characteristics are similar to characteristics of the Australian population, with the exception of age and cultural & language diversity (https://quickstats.censusdata.abs.gov.au/census_services/getproduct/census/2016/quickstat/4?opendocument). The population in South Australia are older (median age: 40 years (South Australian population); 38 years (Australian population)) and less culturally diverse (percentage of people were born in Australia: 71.1% (South Australian population); 66.7% (Australian population)).

A limitation has been added to the “Discussion” section.

The study sample is representative of the South Australian population, and may not be generalised to the whole Australian population which is slightly younger (median age: 38 years (Australian population); 40 years (South Australian population)) and more culturally diverse (percentage of people were born in Australia: 66.7% (Australian population); 71.1% (South Australian population)) [38].

  • the statistical method/techniques used to process the data be adequate to this task (as to this point, I am not that confident that a logistic regression be the more adequate choice; for example, given the sample used, maybe some bayesian techniques would be more favorite on my side, like those employed in these two studies:                        Reopening Italy’s schools in September 2020: a Bayesian estimation of the change in the growth rate of new SARS-CoV-2 cases, BMJ Open http://dx.doi.org/10.1136/bmjopen-2021-051458; 

Modeling COVID-19 pandemic using Bayesian analysis with application to Slovene data, Mathematica Bioscieces, 2020, https://doi.org/10.1016/j.mbs.2020.108466

Response: Thanks so much for the reviewer’s suggestion. In a Vietnamese survey study, a Bayesian regression model was used to identify the factors affecting the respondents’ intention to vaccinate (Khuc 2021). Their sample size was 332. Bayesian method has gained popularity in recent years because it does not require large sample sizes or strict assumptions, as the frequentist approach does (Van de Schoot 2014). However, the sample size of our study was relatively large (n=3003) and all survey data were weighted to be representative of South Australian population. As the reviewer suggested, we applied Bayesian logistic regression to identify the factors associated with vaccine uptake however, the point estimates and intervals do not vary substantially. Please see the results below. We believe the traditional logistic regression with sampling to be robust considering our relatively large sample size and proper weighting process.

Multivariate logistic regression

Bayesian logistic regression

aOdds Ratio

95% confidence interval

aOdds Ratio

95% credible interval

Age

18-29yrs

Ref

Ref

30-49yrs

1.75

1.16-2.65

1.76

1.30-2.40

50-69yrs

3.89

2.68-5.65

4.49

3.68-5.49

≥70yrs

13.55

8.85-20.75

12.57

10.30-15.23

Gender

Male

Ref

Ref

Female

1.09

0.89-1.33

1.09

0.94-1.26

Aboriginal status

No

Ref

Ref

Yes

1.01

0.39-2.60

0.42

0.28-0.65

SEIFA

lowest quintile

Ref

Ref

low quintile

1.31

0.95-1.80

1.21

0.94-1.60

middle quintile

1.60

1.16-2.21

1.37

1.07-1.79

high quintile

1.19

0.85-1.66

1.17

0.92-1.54

highest quintile

1.76

1.27-2.44

1.50

1.18-1.96

Marital status

Married/de facto

Ref

Ref

Single

0.82

0.67-1.01

0.79

0.66-0.94

Education level

Lower than Year 12 education

Ref

Ref

≥Year 12/ TAFE/ certificate/ diploma

1.09

0.85-1.39

1.15

0.97-1.36

Degree or higher

2.24

1.69-2.97

2.06

1.66-2.59

Employment

Unemployed

Ref

Ref

Employed^

1.36

0.83-2.23

1.31

1.01-1.65

Others^^

1.25

0.75-2.10

1.21

0.94-1.56

COB (Country of Birth)

COB English main language

Ref

Ref

COB Non English speaking

0.72

0.54-0.95

0.93

0.76-1.17

Area of residence ^

Metro. Adelaide

Ref

Ref

SA Country

1.25

0.99-1.57

1.21

1.03-1.40

Chronic medical conditions

No

Ref

Ref

Yes

1.27

1.03-1.57

1.24

1.07-1.51

Parents or caregivers of a child/children aged <16 years

No

Ref

Ref

Yes

0.65

0.50-0.86

0.61

0.47-0.72

Van de Schoot, R.; Kaplan, D.; Denissen, J.; Asendorpf, J.B.; Neyer, F.J.; Van Aken, M.A.. A gentle introduction to Bayesian analysis: Applications to developmental research. Child development,2014, 85(3), 842-860. https://doi.org/10.1111/cdev.12169

Khuc, Q.V.; Nguyen, T.; Nguyen, T.; Pham, L.; Le, D.-T.; Ho, H.-H.; Truong, T.-B.; Tran, Q.-K. Young Adults’ Intentions and Rationales for COVID-19 Vaccination Participation: Evidence from a Student Survey in Ho Chi Minh City, Vietnam. Vaccines 2021, 9, 794. https://doi.org/10.3390/vaccines9070794

Obviously, I am not asking to revoluzionize the entire paper but some more reflections and attention to the issues above.

Response: We thank the reviewer for the valuable comments and suggestions.

Reviewer 2 Report

This paper reports descriptive statistics and analyses of sample survey data on COVID-19 immunisation, willingness to be vaccinated and public support for vaccination strategies in Australia. On a first reading, the paper appears well written and the analyses it reports well done. Here are a couple of items to attend to in a revision.

First, near the end of Section 2.3 Statistical Analysis, it is stated:

"All results presented in the regression analyses were weighted. Raking was used to weight respondents incorporating various population characteristics (gender, age, household size, etc) to more closely reflect the South Austral-
ian population using benchmarks derived from the June 2016 ABS Census data." 

This is a crucial methodological step in the statistical analyses. It would be useful and improve the accessibility and understanding of some readers of Vaccines to elaborate the text description of this step, in particular to include a brief verbal description of "raking" with an appropriate citation of one or more publications on the statistical analysis of sample survey data and weighting adjustments so that readers who are not survey statisticians can understand more fully what is referenced here. 

Second, in the same paragraph of the text, it is stated:

"Univariate and multivariate logistic regression analyses were performed to test association between predictor variables and outcome measures." 

That is okay as stated. The one thing I would recommend in addition to this is that later in the text when the empirical regression analyses are described make clear when that the analyses are of the multivariate logistic regressions that include all covariates that passed the first stage bivariate regression threshold of statistical significance.

Author Response

Reviewer 2:

This paper reports descriptive statistics and analyses of sample survey data on COVID-19 immunisation, willingness to be vaccinated and public support for vaccination strategies in Australia. On a first reading, the paper appears well written and the analyses it reports well done. Here are a couple of items to attend to in a revision.

Response: Thank you.

First, near the end of Section 2.3 Statistical Analysis, it is stated:

"All results presented in the regression analyses were weighted. Raking was used to weight respondents incorporating various population characteristics (gender, age, household size, etc) to more closely reflect the South Australian population using benchmarks derived from the June 2016 ABS Census data." 

This is a crucial methodological step in the statistical analyses. It would be useful and improve the accessibility and understanding of some readers of Vaccines to elaborate the text description of this step, in particular to include a brief verbal description of "raking" with an appropriate citation of one or more publications on the statistical analysis of sample survey data and weighting adjustments so that readers who are not survey statisticians can understand more fully what is referenced here. 

Response: The following information highlighted in RED has been added to the “Materials and Methods” section.

Raking was used to weight respondents incorporating various population characteristics (gender, age, household size, etc) to more closely reflect the South Australian population using benchmarks derived from the June 2016 ABS Census data. With raking, we chose a set of variables (gender, age, area of residence, country of birth, dwelling status, marital status, education level, employment status, household size) where the South Australian population distribution is known, and the raking procedure iteratively adjusts the weight for each survey participant until the sample distribution aligns with the South Australian population [12]. The weighting process ensured our results were representative of the South Australian population as a whole.

  1. Dal Grande E, Chittleborough CR, Campostrini S, Tucker G, Taylor AW. Health Estimates Using Survey Raked-Weighting Techniques in an Australian Population Health Surveillance System. Am J Epidemiol. 2015;182(6):544-56.

Second, in the same paragraph of the text, it is stated:

"Univariate and multivariate logistic regression analyses were performed to test association between predictor variables and outcome measures." 

That is okay as stated. The one thing I would recommend in addition to this is that later in the text when the empirical regression analyses are described make clear when that the analyses are of the multivariate logistic regressions that include all covariates that passed the first stage bivariate regression threshold of statistical significance.

Response: As the reviewer suggested, the text in the “Materials and Methods” section has been revised to be clearer to readers. Thank you.

Univariate and multivariate logistic regression analyses were performed to test association between predictor variables and outcome measures. Any above-mentioned covariates with a p-value ≤0.20 on a univariate analysis of association with an outcome measure, were included in a multivariate model. All multivariate logistic regression models only included covariates that achieved bivariate regression threshold of statistical significance (p-value ≤0.20) in the univariate logistic regress analyses.

Round 2

Reviewer 1 Report

No relevant change since my first comments. This is not a revision.
